# Antioxidant and Leishmanicidal Evaluation of *Pulicaria Inuloides* Root Extracts: A Bioguided Fractionation

**DOI:** 10.3390/pathogens8040201

**Published:** 2019-10-23

**Authors:** Hamza Fadel, Ines Sifaoui, Atteneri López-Arencibia, María Reyes-Batlle, Ignacio A. Jiménez, Jacob Lorenzo-Morales, Nabil Ghedadba, Samir Benayache, José E. Piñero, Isabel L. Bazzocchi

**Affiliations:** 1Unité de recherche Valorisation des Ressources Naturelles, Molécules Bioactives et Analyses Physicochimique et Biologiques, Université Constantine-1, Route d’Ain El Bey, 25 000 Constantine, Algerie; hamzafadmel64@yahoo.co (H.F.); sbenayache@yahoo.com (S.B.); 2Instituto Universitario de Enfermedades Tropicales y Salud Pública de Canarias, Departamento de Obstetricia y Ginecología, Pediatría, Medicina Preventiva y Salud Pública, Toxicología, Medicina Legal y Forense y Parasitología, Universidad de La Laguna, Avda. Astrofısico Fco. Sanchez, S/N, 38203 La Laguna, Tenerife, Canary Islands, Spain; ines.sifaoui@hotmail.com (I.S.); atteneri@hotmail.com (A.L.-A.); mreyesba@ull.edu.es (M.R.-B.); jmlorenz@ull.edu.es (J.L.-M.); 3Laboratoire Materiaux-Molecules et Applications, IPEST, University of Carthage, 2070 La Marsa, Tunisia; 4Instituto Universitario de Bio-Orgánica Antonio González, Departamento de Química Orgánica, Universidad de La Laguna, Avenida Astrofísico Francisco Sánchez 2, 38206 La Laguna, Spain; ignadiaz@ull.es; 5Laboratory of Biotechnology of the Bioactive Molecules and Cellular Physiopathology, Department of Biology, University of Batna, 05000 Batna, Algeria; viamessi@gmail.com

**Keywords:** *Leishmania*, antioxidant, *Pulicaria*, bioguided, therapy

## Abstract

Leishmaniasis remains a major world health problem, and in particular, Algeria ranks second for the incidence of cutaneous leishmaniasis. *Pulicaria inuloides* is a well-known Arabian Peninsula medicinal plant. In the present study, the chloroform, ethyl acetate and *n*-butanol extracts from the roots of *Pulicaria inuloides* were analyzed for antioxidant activity and its correlation with the total phenolic and flavonoid contents. The highest antioxidant activity using a DPPH assay was showed by the ethyl acetate extract (IC_50_ 4.08 µg/mL), which also had the highest total phenolic content (307.12 µgAGE). Furthermore, *P. inuloides* root extracts were evaluated against *Leishmania amazonensis* and *Leishmania donovani*. The results highlighted the chloroform extract as the most active one against both tested *Leishmania* strains. A bioguided fractionation of the chloroform extract led to the isolation of (8*R*:8*S*)-(75:25 er)-10-isobutyryloxy-8,9-epoxy-thymol isobutyrate as the main bioactive component, showing a potent leishmanicidal activity on *L. amazonensis* promatigote and amastigote stages (IC_50_ 5.03 and 2.87 µM, respectively) and a good selectivity index on murine macrophages (CC_50_ 19.37 µM). This study provides the first report of the antioxidant and leishmanicidal activities of *P. inuloides* root extracts and the results point to this species as a source of potential bioactive agents.

## 1. Introduction

Leishmaniasis, a neglected tropical disease caused by protozoan parasites belonging to the genus *Leishmania*, remains a major world health problem [1]. In particular, Algeria is endemic for cutaneous leishmaniasis (CL) and visceral leishmaniasis (VL) [2]. CL, also called “Biskra boil” in the local language, is a serious public health problem because the country has the second largest focus in the world after Afghanistan. It is reported to be endemic in 328 third sub-national administrative levels with 10 million populations at risk, rising 13106 numbers of cases reported in 2017 [3]. Regarding VL, the cases have been detected mainly from the central and eastern parts of the Tell region, and new foci have appeared and the number of cases has resurged, with 34 new cases reported in 2017 [4]. The Leishmaniasis National Control Program (LNCP) in Algeria was established in 2006, with a vector control program based on pyrethroids as insecticides, but without reservoir host control program [5]. The treatment of the population is provided free of charge in the public sector, with the following antileishmanial medicines included in the national List of Essential Medicines: amphotericin B deoxycholate, liposomal amphotericin B and meglumine antimoniate [6,7]. In spite of the high prevalence, current chemotherapy for leishmaniasis is compromised by high cost, toxicity associated with long-term treatments, route of administration, drug resistance, and different strain sensitivity to the available drugs [8]. These drawbacks lead to an urgent need to develop new treatments with acceptable efficacy and safety profile. In this regard, plants have demonstrated to be an important source of leishmanicidal drugs owing to their accessibility, structural diversity, low cost and possible rapid biodegradation [9].

The genus *Pulicaria* belonging to the Asteraceae family (tribe Inuleae) consists of about 100 species distributed in Europe, North Africa and Asia, particularly around the Mediterranean region [10,11]. Some *Pulicaria* species are used as traditional medicines for treatment of a variety of illnesses due to their anticancer, antioxidant, antimicrobial, antifungal and antimalarial properties, among others [11,12,13]. *Pulicaria inuloides* (Poir.) DC is a well-known Yemeni medicinal plant. The leaves are claimed to have antibacterial and antioxidant properties, and are also used to flavor foods and to make an herbal tea [14]. The chemical composition [14,15,16] and medicinal uses as anticancer, antibacterial and antioxidant [14,17] of the *P. inuloides* essential oil obtained by hydro-destillation of the leaves and flowers have been reported. Moreover, the antioxidant properties and total phenol and flavonoid contents of different extracts from the leaves of *P. inuloides* were analyzed by Al-Hajj and co-workers [15]. In addition, the isolation of *ent*-kaurane diterpenoids and flavonoids from the aerial part of *P. inuloides* was reported, along with their antimicrobial, cytotoxic and antioxidant evaluation [18]. 

Our previous works reported the in vitro antiprotozoal evaluation of extracts from the aerial part of *P. inuloides*, and the characterization of quercetagetin-3,5,7,3’-tetramethyl ether as the main component of the active CHCl_3_ extract [19]. In the present work, and as a continuation of the *P. inuloides* medicinal plant research, the antioxidant activity, total phenolic and flavonoid contents, and leishmanicidal properties of *P. inuloides* roots are reported. The bioassay-guided fractionation of the active CHCl_3_ extract led to the isolation and characterization of 10-isobutyryloxy-8,9-epoxythymol isobutyrate as the main leishmanicidal component, highlighting this plant as a promising source of antiparasitic agents.

## 2. Results and Discussion

### 2.1. DPPH Radical Scavenging Activity 

Oxidative stress has been associated with the development and progression of several diseases, including cancer, inflammation, atherosclerosis, cardiovascular diseases or diabetes. Thus, administration of exogenous antioxidants is a promising way of combating the undesirable effects of reactive oxygen species (ROS) induced oxidative damage. In this regard, plants have demonstrated to be an important source of antioxidants such as polyphenols or flavonoids [20]. Moreover, taking into consideration that some *Pulicaria* species, and in particular *P. inuloides*, are claimed to have antioxidant properties [11], the antioxidant activity of *P. inuloides* roots was analyzed.

The results of the antioxidant activity (Table 1, Figure 1) by the DPPH radical scavenging assay of three extracts from the roots of *P. inuloides* indicated strong ability of the ethyl acetate extract to act as DPPH scavenger, with an IC_50_ value of 4.08 µg/mL, followed by the *n*-butanol and chloroform extracts (8.76 and 12.85 µg/mL, respectively). The antioxidant activity, in particular this showed by the ethyl acetate extract is significantly high when compared to the standard ascorbic acid (1.30 µg/mL). Therefore, to investigate the correlation between the antioxidant activity and presence of antioxidants compounds into the different extracts, the total phenolic and flavonoid contents were determined.

### 2.2. Total Phenolic and Flavonoids Contents of Extracts 

The amount of total phenols in the extracts was measured by the Foline-Ciocalteau method [21] in terms of gallic acid equivalent (GAE) from the calibration curve of gallic acid standard solution, using the equation: y = 0.004x + 0.017 with R² = 0.988, where y = absorbance at 765 nm and x = concentration of total phenolic content (Appendix A). The ethyl acetate and *n*-butanol extracts showed the highest values (307.12 and 229.76 µgGAE/mg, respectively), whereas the chloroform extract showed a low total phenol content (32.28 µgGAE/mg) as seen in Table 1. These results indicated a direct correlation between the phenols content and the radical-scavenging capacity of the extracts. 

Previous studies have reported the total phenolic content of *P. inuloides* essential oil (144 mgGAE/g) [14] and those of the methanolic, ethanolic and diethyl ether extracts of *P. inuloides* leaves (91.2, 89.9 and 64.9 mgGAE/g, respectively) [15]. Our results indicated that both, the ethyl acetate and *n*-butanol extracts from the roots of the plant have a higher total phenolic content than those for the essential oil and leaves extracts and by instant a higher antioxidant capacity. The total flavonoid content was calculated from a calibration curve (Appendix A), and the results were expressed as µg quercetin equivalent per mg of extract (µgQE/mg). The total flavonoid content of the extracts was measured using the QE equation: y = 0.027x + 0.014 with R² = 0.999, where y = absorbance at 420 nm, and x = concentration of total flavonoid content. The results indicated low flavonoids content in the ethyl acetate and *n*-butanol extracts (5.08 and 6.31 µgQE/mg, respectively), while this type of compounds could not be detected in the chloroform extract (Table 1). A correlation between the total phenolic content and the antioxidant capacity of the extracts was observed. Previous reported studies revealed that phenolic compounds are major antioxidant constituents in the leaves of *P. inuloides* extracts [15], and the good correlation observed in our study support the hypothesis that phenolic compounds in *P. inuloides* roots contribute significantly to their antioxidant properties.

### 2.3. Bioassay-Guided Fractionation 

The roots of *P. inuloides* (200 g) were extracted with 80% ethanol, and the dry residue further suspended in water and extracted, successively, using three organic solvents to yield CHCl_3_, EtOAc and *n*-BuOH extracts. The highest extraction yield was obtained with *n*-BuOH (2%), followed by extraction with EtOAc (0.50%) and CHCl_3_ (0.44%). The three organic extracts were submitted to an in vitro leishmanicidal screening (Table 2, Scheme 1), revealing that the chloroform extract was the most active against *L. amazonensis* and *L. donovani* promastigote form. Therefore, the chloroform extract (0.89 g) was subjected to silica gel column chromatography, eluted with dichloromethane-methanol of increasing polarity to afford two main fractions: F1 (268.0 mg) and F2 (340.0 mg). The active fraction F1 was further chromatographed, using column chromatography on silica gel eluted with hexanes-ethyl acetate of increasing polarity to obtain three fractions: F1A (25.0 mg), F1B (131.0 mg), and F1C (13.0 mg). 

The most active fraction F1B against both *Leishmania* strains (Table 3) was subjected to preparative thin layer chromatography (PTLC) to afford three sub-fractions (F1B1-F1B3) (Scheme 1). Sub-fractions F1B1 and F1B2 were inactive at 200 µg/mL on *L. amazonenis* and *L. donovani*. On the other hand, sub-fraction F1B3 showed a potent leishmanicidal effect with IC_50_ values of 5.03 ± 0.29 µM and 4.65 ± 0.10 µM on both *Leishmania* strains, respectively, even somewhat higher on *L. amazonensis* that Miltefosine used as the reference drug (IC_50_ values of 6.48 µM and 3.32 µM on *L. amazonensis* and *L. donovani*, respectively, Table 4 and Figure 2). Moreover, evaluation of this active sub-fraction on murine macrophages (CC_50_ 19.37 µM ± 0.23 µg/mL) in the search for selectivity indicated a moderate selective index (SI), with values of 3.9 and 4.2 for these *Leishmania* strains, respectively [22].

Activity assays against *L. amazonensis* intracellular amastigotes (Table 4) revealed that F1B3 was also more potent (IC_50_ 2.87 µM) that the reference drug, miltefosine (IC_50_ 3.12 µM), which increases the interest of this active sub-fraction as potential leishmanicidal agent. 

TLC analysis of fraction F1B3 suggested it could be a pure compound (Figure 3). Therefore, spectrometric and spectroscopic studies, including 1D- and 2D-NMR spectra (Appendix A), led to the characterization of F1B3 as 10-isobutyryloxy-8,9-epoxy-thymol isobutyrate (1) [23] as an enantiomeric mixture with around a 75:25 scalemic proportion as determined by the integration of the signals in its ^1^HNMR spectrum. Previously, the absolute configuration and scalemic proportion determination of epoxy-thymol derivatives using (*S*)-BINOL as a chiral solvating agent combined with vibrational circular dichroism (VCD) studies has been reported [24]. By applying this methodology the authors determined the enantiomeric ratio of this expoxy-thymol derivative as (8*S*:8*R*)-(75:25 er) with specific rotation value [α]_436_ +18.9. Compound 1 shows a specific rotation of [α]^20^_D_ -18.5 value, suggesting its structure corresponds to (8*R*:8*S*)-(75:25 er)-10-isobutyryloxy-8,9-epoxy-thymol isobutyrate (1).

Compound 1 has been isolated from 40 different plants that represent 60% of the total reported vegetal species containing epoxythymol derivatives. Only about 10% of the known functionalized thymol derivatives have been evaluated as antibacterial, anti-inflammatory, antioxidant, antiprotozoal, cytotoxic, piscicidal, or allelophatic agents [25]. Regarding epoxythymol derivatives, evaluation on *Entamoeba histolytica* and *Giardia lamblia* of thymol isobutyrate derivatives revealed a moderate antiprotozoal activity on both protozoa [26]. However, studies on their leishmanicidal or antioxidant activities have not been reported.

To our knowledge, this is the first report of thymol derivative 1 in *Pulicaria* genus. Previously, the leishmanicidal bioguided fractionation of the active chloroform extract from the aerial part of *P. inuloides* led to the isolation of quercetagetin-3,5,7,3’-tetramethyl ether as the main bioactive component, exhibiting a moderate leishmanicidal activity with an IC_50_ value of 0.483 mM on *L. amazonensis* [19]. Furthermore, the present study reveals that a thymol derivative, the major component from the roots of this species, is responsible for the leishmanicidal activity. This further supports *P. inuloides* as a source of promising leishmanicidal agents.

## 3. Materials and Methods

### 3.1. General Experimental Procedure 

Optical rotation was measured on a Perkin Elmer 241 automatic polarimeter in CHCl_3_ at 20 °C. ^1^H (600 MHz), ^13^C (150 MHz) Nuclear Magnetic Resonance (NMR) spectra were recorded on a Bruker Avance 600 spectrometer, the chemical shifts are given in δ (ppm) with residual CDCl_3_ (δ_H_ 7.26, δ_C_ 77.0) as internal reference and coupling constants in Hz; the experiments were carried out with the pulse sequences given by Bruker. EIMS and HREIMS were collected with a Micromass Autospec spectrometer. Silica gel 60 (particle size 15–40 and 63–200 μm, Macherey-Nagel) was used for column chromatography, while silica gel 60 F254 (Macherey-Nagel) were used for analytical or preparative TLC. The spots were visualized by UV light and heating silica gel plates sprayed with H_2_O-H_2_SO_4_-AcOH (1:4:20). All solvents used were purchased from Sigma Aldrich.

### 3.2. Plant Material and Extraction

*Pulicaria inuloides* (Poir.) DC. was collected at flowering stage in 2013 in Bechar Dam Djorf Ettorba (80 km south west from Bechar) by Professor Samir Benayache. A voucher specimen (PU/105/VAR/05-15) was identified by Ben Abd El-Hakem Mohamed, Head of the Protection Plants Service in Algeria and was deposited in the Herbarium of the VARENBIOMOL Research Unit, Université des Frères Mentouri, Constantine, Algeria.

The powdered air-dried roots of *P. inuloides* (200 g) were extracted three times with 80% ethanol (3 × 1 L) at 26 °C. The extracts were filtered through cotton and solvent was evaporated under reduced pressure to dryness. The dry residue (45 g) was further suspended in water and extracted, successively, with chloroform (3 × 100 mL), ethyl acetate (3 × 100 mL) and *n*-butanol (5 × 100 mL) at room temperature. Each extract was collected, separately, and concentrated by a rotary vacuum evaporator to remove the solvent to yield the CHCl_3_ (0.89 g, 0.44% *w/w* on dry plant material), EtOAc (1 g, 0.50% *w/w*) and *n*-BuOH (4 g, 2.0% *w/w*) extracts. 

### 3.3. DPPH Radical Scavenging Assay 

The antioxidant activity of the extracts has been evaluated using the DPPH method (2,2-diphenyl-1-picrylhydrazyl) [27]. All extracts have been prepared in methanol at a concentration of 8 mg/mL. Then, 30 μL of sample solutions at different concentrations were added to 3 mL of DPPH solution (0.04 mg/mL). After 30 min of incubation in the dark at room temperature, the absorbance of the samples was measured at 517 nm with an UV-Vis spectrophotometer (Thermo Scientific Evolution 300). The capability to scavenge the DPPH radical was calculated using the following equation: % inhibition = [(Ac – As)/Ac] x 100; where Ac is the blank control absorbance and is the sample absorbance at 517 nm. Ascorbic acid was used as a positive control. All determinations were carried out in duplicate, and the results were expressed as IC_50_ values (concentration of sample required to scavenge 50% DPPH free radicals) calculated from a calibration curve using Microsoft Excel [28]. 

### 3.4. Determination of Total Phenolic Content 

The total phenolic content of extracts was measured by the Folin-Ciocalteu method [21] with slight modifications. Briefly, the extracts were prepared at a concentration of 1.0 mg/mL in distilled water. There were 500 μL aliquot transferred into a test tube and 1.0 mL of Folin-Ciocalteu reagent (1 N) was added. The mixture was allowed to stand at room temperature for 4 min, and afterward, 5 mL of 20% sodium carbonate were added to the mixture and was kept at room temperature for 2 h prior to recording the absorbance at 765 nm with an UV-Vis spectrophotometer. All determinations were carried out in duplicate. The total phenolic content was calculated from a calibration curve, and the results are expressed as µg gallic acid equivalents per mg of extract (µgGAE/mg).

### 3.5. Determination of Total Flavonoid Content

The total flavonoid content in extracts was determined by the aluminium trichloride method [29], using quercetin as the reference compound. Briefly, a volume of 2 mL of ethanolic solution of extract (1 mg/mL) was mixed with 2 mL of 2 % AlCl_3_. After incubation at room temperature for 1 h, the absorbance was measured at 420 nm. All determinations were carried out in duplicate. The total flavonoid content was calculated from a calibration curve, and the results were expressed as µg quercetin equivalent per mg of extract (µgQE/mg). 

### 3.6. Parasite Strains

The leishmanicidal activity was evaluated against promastigotes of *L. amazonensis* (MHOM/BR/77/LTB0016) and *L. donovani* (MHOM/IN/90/GE1F8R). Promastigotes were cultured in Schneider’s medium (Sigma-Aldrich, Madrid, Spain) supplemented with 10% fetal bovine serum at 26 °C and were grown to the log phase as per previous methods. For some of the assays, the parasites were also cultured in RPMI 1640 medium (Gibco), with or without phenol red [30].

### 3.7. In Vitro Effect on Promastigote Forms of Leishmania spp

The activity of crude extracts and fractions were determined by the modified alamarBlue^®^ assay (Invitrogen/Life Technologies, Madrid, Spain) as previously described [31]. This simple and rapid test is based on oxido/reduction reaction. Briefly, the oxidized, blue, non-fluorescent Alamar Blue is reduced to a pink fluorescent dye in the medium by cell activity. This reaction could be measured either by colorimetric or fluorimetric [32]. Samples were dissolved in dimethyl sulfoxide (DMSO) and further dilutions were made in RPMI 1640 medium. The final DMSO concentration never exceeded 0.1% (*v/v*) with no effect on the parasites proliferation or morphology. Promastigotes of *L. amazonensis* and *L. donovani* were grown at 26 °C in RPMI 1640 modified medium (Gibco) and supplemented with 10% heat-inactivated fetal bovine serum. Logarithmic phase cultures were used for experimental purposes, and the in vitro susceptibility assay was performed in sterilized 96-wellmicrotiter plates (Corning™). To these wells, 10^6^/well parasites were added, and the samples at the concentration to be tested. The final volume was 200 μL in each well. After an incubation of 72 h, analysis of the plates was carried out visually using an inverted microscope. Subsequently, in the case of the pure compound, the plates were analyzed on an EnSpire multimode plate reader (PerkinElmer, MA, USA) using a test wavelength of 570 nm and a reference wavelength of 630 nm. Miltefosine, kindly provided by Æterna Zentaris Inc., was used as reference drug. Percentage of inhibition, 50% inhibitory concentrations (IC_50_) for the active samples, was calculated by linear regression analysis with 95% confidence limit. All experiments were performed three times each in duplicate, and the mean values were calculated. A non-parametric regression, adjusting the data to a four-parameter logistic curve was used for analysis of the data. The obtained inhibition curves statistical analysis was undertaken using Sigma Plot 12.0 software program (Systat Software Inc.). 

### 3.8. In Vitro Effect on Leishmania Amazonensis Amastigote Stage

Activity assay against intracellular amastigotes was performed according to Jain et al. (2012) [33]. Macrophages (J774A.1 cell line) were placed in a 96-well flat bottom plate at a density of 2 × 10^5^/mL in RPMI supplemented with 10% SBF and incubated for an hour at 37 °C in 5 % CO_2_ environment. Additionally, 100 µL of stationary phase promastigotes (7 days old culture) were added in a 10:1 ratio, and plates were re-incubated at 37 °C overnight to allow a maximum infection. After incubation, free promastigotes were washed off with the same medium at least 3 times. 50 μL of culture medium (RPMI-1640 with 10% FBS) were added into each well. Subsequently, a serially dilution of test compounds was made in a 96-deep well plate with the same medium, and then 50 μL of this serially-diluted standard were added to each well. The plates were incubated at 37 °C, 5% CO_2_ for 24 h. After incubation, the medium from each well was removed, and 30 μL of Schneider (with 0.05% SDS) was added to each well. Plate was shacked for 30 sec. and 170 μL of Schneider medium were added to each well. AlamarBlue at 10% was added into each well of the 96-well plates and incubated at 26 °C for 72 h to allow transformation of rescued amastigotes to promastigotes. After incubation, the emitted fluorescence was measured in a Perkin Elmer EnSpire spectrofluorometer at 585 nm.

### 3.9. Cytotoxicity Assay 

Cytotoxicity was evaluated after 24 h incubation of macrophages J774A.1 (ATCC TIB-67) murine macrophage cell line with different concentrations of samples. The viability of the macrophages was determined with alamarBlue^®^ assay as previously described [34]. Briefly, the macrophages at concentration of 10^5^ cells/mL cultured in RPMI medium supplemented with 10% fetal bovine serum at 37 °C in a 5% CO_2_ were incubated with different concentrations of sample and AlamarBlue reagent is added at 10% of the final volume. After 24 h of incubation, the plates were analyzed on an EnSpire multimode plate reader (PerkinElmer, MA, USA) using a test wavelength of 570 nm and a reference wavelength of 630 nm. Dose response curves were plotted and the CC_50_ was obtained. The analyses were performed in triplicate.

### 3.10. Leishmanicidal Bioguided Fractionation 

Fractionation of the active chloroform extract was performed against the promastigote stage of *L. amazonensis* and *L. donovani* (Scheme 1). The CHCl_3_ extract (0.89 g) was subjected to column chromatography on silica gel and eluted with dichloromethane-methanol of increasing polarity (CH_2_Cl_2_-MeOH; 100:0, 70:30, 30:70, 0:100). Four fractions of 50 mL were collected, and then combined in two fractions on the basis of their thin layer chromatography (TLC) profiles. Based on the results of leishmanicidal screening assay, the active fraction F1 (268 mg) was submitted to further silica gel column chromatography eluted with hexanes-ethyl acetate of increasing polarity (100:0 to 0:100). Three sub-fractions (F1A-F1C) were obtained based on their TLC profile, and tested for their activity against *L. amazonensis* and *L. donovani*. The active F1B fraction (36 mg) was purified on a preparative silica gel thin layer chromatography (PTLC), using mixtures of hexanes-ethyl acetate (50:50) to obtain sub-fractions F1B1-F1B3. Sub-fraction F1B3 yielded compound 1, as a pale yellow amorphous solid (10 mg), which was identified as 10-isobutyryloxy-8,9-epoxy-thymol isobutyrate by comparison of its spectroscopic and spectrometric data with those previously reported [23,24].

## 4. Conclusions

This study aimed to investigate the therapeutic potential of *P. inuloides*, an Algerian species used for traditional medicine. The results reported herein indicated that the ethyl acetate extract of the roots possesses a significant antioxidant profile, in accordance with a high total phenolic content. Moreover, the leishmanicidal bioguided fractionation against *L. amazonensis* and *L. donovani* of the active chloroform extract led to the isolation of a thymol derivative as the main bioactive component, showing higher potency that the reference drug against promastigote and amastigote forms of *L. amazonensis*. This study, and the previously reported one on the aerial part of this species, underlines this plant as being a potential source of therapeutic agents. Further studies will be conducted in intracellular amastigote form of the parasites to explore the potential of *P. inuloides* and its metabolites as alternative or complementary leishmanicidal remedies.

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
