# Peer review of "Antioxidant and Leishmanicidal Evaluation of Pulicaria Inuloides Root Extracts: A Bioguided Fractionation"

_pathogens, 2019, doi:10.3390/pathogens8040201_

Round 1

Reviewer 1 Report

The manuscript evaluated the antioxidant and anti-leishmania activity of root extracts from Pulicaria inuloides with the aim to discover anti-leishmanial drugs against cutaneous leishmaniasis in the Arabian Peninsula. Anti-protozoal activity of extracts of the aerial parts of this plant was published previously by the same authors (2018). In the present publication authors claim the identification of components with radical-scavenging activity and anti-leishmania activity in root extracts. Study of natural extracts and identification of the compounds with anti-leishmania activity and antioxidant capacity is interesting. However, the manuscript lacks hard data to demonstrate association of antioxidant capacity of the fractions to anti-leishmania capacity. No simple positive correlation between these two variables was calculated.

Authors claim that the compounds identified are potential anti-leishmania candidates. However, the most active sub fractions had non impressive IC50 values (4.6-5uM) and were toxic to host cells. Resulting in poor selectivity index (SI) of 3.9 and 4.2. SI values are lower than the 20s obtained in the compounds identified in their previous publication and lower that standard acceptable therapeutic index, 10.

Major: pros and cons of compounds identified as anti-leishmania drug candidates should be addressed more clearly with reference to compounds found, by the same authors, in the same plant.

Points that should be addressed in results section:

Authors should include results to support the positive correlation of anti-leishmania capacity and antioxidant potency of the same fractions to support main conclusion. Anti-leishmania activity correlated to anti-oxidant capacity should be experimentally demonstrated.

IC50 curves showing qualitative data of anti-leishmania activity against the control fractions would be shown and are more relevant than qualitative data shown in tables 2 and 3. Vehicle controls need to be included in each standard curve.

Qualitative scale in table is not explained.

Anti-leishmania activity of control fractions (vehicle), which are potentially toxic per se, are missing from the tables and must be included.

Author Response

Authors claim that the compounds identified are potential anti-leishmania candidates. However, the most active sub fractions had non impressive IC50 values (4.6-5uM) and were toxic to host cells. Resulting in poor selectivity index (SI) of 3.9 and 4.2. SI values are lower than the 20s obtained in the compounds identified in their previous publication and lower that standard acceptable therapeutic index, 10.

The most active sub fraction, F1B3, showed to be a main compound. It showed an IC50 value of 5.03 µM against L. amazonensis, whereas miltefosine, used as reference drug, showed an IC50 value of 6.48 µM. Thus, although the reviewer indicated F1B3 had non an impressive IC50 value, we consider F1B3 exhibited a potent leishmacidal activity, higher than miltefosine used as first-line drug for leishmanisis treatment. Moreover, F1B3 also showed a close IC50 value to the reference drug (4.65 µM versus 3.32 µM) against L. donovani.

Regarding the selectivity index (SI), the reviewer is right concerning the compound identified as leishmanicidal from the aerial part of the plant under study showed a higher SI than compound 1. However, we consider that a SI around 4 could not toxic to host cells. In fact, according to Likhitwitayawuid et al. (1993), molecules used in therapy should present a SI higher than 1 (Likhitwitayawuid K, Angerhofer CK, Chai H, Pezzuto JM, Cordell GA, Ruangrungsin, 1993) Cytotoxic and antimalarial alkaloids from the tubers of Stephania pierrei. J. Nat. Prod. 56(9):14681478). This reference has been included in the revised manuscript. Even though, we corrected the text in the revised manuscript (pg 4, line 156): “..indicated a moderate selective index (SI), with values of 3.9 and 4.2 for these Leishmania strains, respectively”.

Moreover, our study reveals the potential of P. inuloides as source of therapeutic agents. As we underline in the Conclusions section “Further studies will be conducted in intracellular amastigote form of the parasites to explore the potential of P. inuloides and its metabolites as alternative or complementary leishmanicidal remedies”.

Major: pros and cons of compounds identified as anti-leishmania drug candidates should be addressed more clearly with reference to compounds found, by the same authors, in the same plant.

Regarding this comment, our previous published work on P. inuloides was focused on the areal part of the plant, and at this time queretagetin-3,5,5,7,3-ether was identified as the main active metabolite. In the present work, the leismanicidal potential of the roots of the species under study is reported. This fact is discussed in the Introduction section, but also in Results and Discussion section in the present manuscript.

On pg 2: “Our previous works reported the in vitro antiprotozoal evaluation of extracts from the aerial part of P. inuloides, and the characterization of quercetagetin-3,5,7,3'-tetramethyl ether as the main component of the active CHCl3 extract”

On pg 4: “Previously, the leishmanicidal bioguided fractionation of the active chloroform extract from the aerial part of P. inuloides led to the isolation of quercetagetin-3,5,7,3'-tetramethyl ether as the main bioactive component, exhibiting a moderate leishmanicidal activity with an IC50 value of 0.483 mM on L. amazonensis.”

Points that should be addressed in results section:

Authors should include results to support the positive correlation of anti-leishmania capacity and antioxidant potency of the same fractions to support main conclusion. Anti-leishmania activity correlated to anti-oxidant capacity should be experimentally demonstrated.

The aim of the work was not to found a correlation between anti-Leishmania capacity and antioxidant potency. In fact, the most antioxidant activity was showed by the ethyl acetate extract (significantly high when compared to the standard ascorbic acid), whereas the higher leishmanicidal activity was found in the CHCl3 extract. Therefore, both properties are not correlated, but it is of interest that the studied species could has several medicinal properties.

We study the antioxidant properties of the plant and correlated with the total phenolic and flavonoid contents, because it is widely known the interest of antioxidants as a promising way of combating the induced oxidative damage, and because P. inuloides is claimed to possess antioxidant properties.

On pg 3: “Therefore, to investigate the correlation between the antioxidant activity and presence of antioxidants compounds into the different extracts, the total phenolic and flavonoid contents were determined.”

Moreover, Algeria is endemic for cutaneous and visceral leishmaniasis, which is a serious public health problem, leading to an urgent need to develop new treatments with acceptable efficacy and safety profile. Plants have demonstrated to be an important source of leishmanicidal drugs owing to their accessibility, structural diversity, low cost and possible rapid biodegradation. This fact encouraged us to investigate the leishmanicidal potential of the roots of P. inuloides, a plant that grows in Algeria.

IC50 curves showing qualitative data of anti-leishmania activity against the control fractions would be shown and are more relevant than qualitative data shown in tables 2 and 3. Vehicle controls need to be included in each standard curve.

The first step in this work was the screening of the extracts (CHCl3, EtOAc and n-BuOH) by microscopic observations (shape, mobility of parasites) to select the most active one. The most active extract was further submitted to fractionation to get in this case 2 fractions (F1 and F2) which were also evaluated following the same methodology. Sub-fractions F1A-F1C were also screening for activity. Thus, qualitative data were obtained for the extracts and fractions evaluated. The IC50 values were only calculated for the pure compound. This point was clarified in Materials and Methods section (pg 8). Even though, qualitative scale in Tables 2 and 3 is explained in the revised manuscript.

Regarding the vehicle control, as indicated in Materials and Methods, “the final DMSO concentration never exceeded 0.1% (v/v) with no effect on the parasites proliferation or morphology”

Qualitative scale in table is not explained.

As suggested by the reviewer, qualitative scale in Tables 2 and 3 is explained in the revised manuscript as follows:

“Key: (-) inactive (viability > 80%), (+) low activity (viability 60-80%), (++) moderate activity (viability 20-60%), (+++) potent activity (viability < 20%).”

Anti-leishmania activity of control fractions (vehicle), which are potentially toxic per se, are missing from the tables and must be included.

As indicated before, the vehicle control (DMSO), as indicated in material and methods, was used at a concentration that never exceeded 0.1% (v/v), with no effect on the parasites proliferation or morphology.

Reviewer 2 Report

In this paper, Fadel et al described the leishmanicidal bioguided fractionation of Pulicaria inuloides root and identified the anti-Leishmania activity of a thymol derivative. Taking into account the lack of efficient vaccines and the high cost, toxicity, parenteral route of administration and resistance to currently available anti-Leishmania drugs, search of novel molecules is critical. The article is easy to read, straightforward, and addresses an important and current topic.

Major comment

The methodology used to study the leishmanicidal activity of the different fractions and sub-fractions of Pulicaria inuloides root, and its cytotoxicity seems to be somewhat brief and the bibliographic reference indicated does not help to clarify the materials and methods.

Specific comments:

Introduction

The first paragraph refers to the epidemiology of leishmaniosis in Algeria, revealing some concern about the actual situation. In view of this, the choice of the tested Leishmania species is surprising, as two non-endemic species were tested.

Materials and methods

Line 243- “Samples were dissolved in dimethyl sulfoxide (DMSO) and further dilutions were made in RPMI 1640 medium.” What was the final concentration of DMSO in the wells?

Line 248- “To these wells, 106/well parasites were added, and the samples at the concentration to be tested.” I would like to know if controls were used. What were the controls? Miltesosine was used in this experience as standard drug?  At line 156 (Results and Discussion) miltefosine IC 50 values for both Leishmania strains were presented, but in materials and methods there is no reference to controls.

Line 249- “After an incubation of 72 h, analysis of the plate was carried out visually using an inverted microscope. Subsequently, the plates were analysed for 72 h on an EnSpire multimode plate reader (PerkinElmer, MA, USA) using a test wavelength of 570 nm and a reference wavelength of 630 nm.” This phrase seems to indicate that at 72 hours the cultures were observed and subsequently incubated for a further 72 hours and then read. If this is the case, is not a 6 days incubation excessive for parasites? Why such a long incubation period of the parasite with the compounds?

Line 255- “A paired two-tailed t test was used for analysis of the data.”- was the normality of the values tested? It wouldn't be more convenient to use a nonparametric test?

Line 258- About cytotoxicity assay, did you use controls? What were the controls?

Results and Discussion

Line 98- I think you can aesthetically improve the Figure 1 by removing the square that externally limits the image as well as the horizontal lines.

Line 107- “…concentration of total phenolic content (Supplementary Materials, Figure S1) . The ethyl acetate…”. Remove space, please!

Lines 140 and 156- Can you explain exactly what means (-) inactive, (+) low activity, (++) moderate activity, (+++) potent activity? As Alamar Blue® Assay is a quantitative colorimetric assay, can you give the quantitative values?

153- “…amazonensis and L. donovani, respectively) Moreover, evaluation of this active sub-fraction on murine 153”. Missing punctuation mark.

Conclusions

Leishmanicidal activity of the thymol derivative was tested on promastigotes that are not the form of the parasite that infect the host cell. However, promastigotes are not the form of the parasite that infects the definitive host. I think it is important to mention this point in the conclusions, stressing that this is a preliminary study. In terms of future prospects, it will be important to explore the potential of P. inuloides and its metabolites in the intracellular amastigote form of the parasite, especially autochthonous Leishmania species.

Author Response

Major comment

The methodology used to study the leishmanicidal activity of the different fractions and sub-fractions of Pulicaria inuloides root, and its cytotoxicity seems to be somewhat brief and the bibliographic reference indicated does not help to clarify the materials and methods.

As suggested by the reviewer, the methodology used to test leishmanicidal and cytotoxicity activities have been extended in the Materials and Methods section, and a bibliographic reference added to the revised manuscript.

Specific comments:

Introduction

The first paragraph refers to the epidemiology of leishmaniosis in Algeria, revealing some concern about the actual situation. In view of this, the choice of the tested Leishmania species is surprising, as two non-endemic species were tested.

In North Africa, the Maghreb area is endemic for L. infantum/chagasi as well as for L. major, where Algeria is located. There are countries from the same area, as Sudan, where L. donovani cases are also common. In this fact, we could adduce that inside the L. donovani complex, causative of visceral leishmaniasis, are L. infantum, L. chagasi and L. donovani species, taxonomically very near between them. In the laboratory where the activity assays were carried out, L. donovani parasites are maintained normally, and been near taxonomically to the one present in Algeria, was the chosen strain to perform the experiments.

A reference was included in the Introduction of the revised manuscript to record this statement: Aronson N et al 2017: Diagnosis and Treatment of Leishmaniasis: Clinical Practice Guidelines by the Infectious Diseases Society of America (IDSA) and the American Society of Tropical Medicine and Hygiene (ASTMH). Am J Trop Med Hyg. 11; 96 (1): 24-45. doi: 10.4269/ajtmh.16-84256).

In addition, L. amazonensis strain was used in order to test the extracts/fractions and active compound against a causative strain of the cutenous leishmaniasis.

Materials and methods

Line 243- “Samples were dissolved in dimethyl sulfoxide (DMSO) and further dilutions were made in RPMI 1640 medium.” What was the final concentration of DMSO in the wells?

Regarding this point, the following sentence was includes in the revised version of the manuscript (pg 7, line 255):

“The final DMSO concentration never exceeded 0.1% (v/v) with no effect on the parasites proliferation or morphology”

Line 248- “To these wells, 106/well parasites were added, and the samples at the concentration to be tested.” I would like to know if controls were used. What were the controls? Miltesosine was used in this experience as standard drug?  At line 156 (Results and Discussion) miltefosine IC 50 values for both Leishmania strains were presented, but in materials and methods there is no reference to controls.

The reviewer is right, and a sentence was added in Materials and Methods of the revised manuscript regarding this item (pg 7, line 265).

“Miltefosine, kindly provided by Æterna Zentaris Inc., was used as reference drug”.

Line 249- “After an incubation of 72 h, analysis of the plate was carried out visually using an inverted microscope. Subsequently, the plates were analysed for 72 h on an EnSpire multimode plate reader (PerkinElmer, MA, USA) using a test wavelength of 570 nm and a reference wavelength of 630 nm.” This phrase seems to indicate that at 72 hours the cultures were observed and subsequently incubated for a further 72 hours and then read. If this is the case, is not a 6 days incubation excessive for parasites? Why such a long incubation period of the parasite with the compounds?

The reviewer is right, and the incubation period of 72 h was repeated by error. Thus, this point was clarified in the revised manuscript and “for 72 h” on pg 8, was removed.

Line 255- “A paired two-tailed t test was used for analysis of the data.”- was the normality of the values tested? It wouldn't be more convenient to use a nonparametric test?

The reviewer is right, this sentence was removed “A paired two-tailed t test was used for analysis of the data. Values of p < 0.05 were considered significant.”

Yes, a non-parametric regression was used for analysis of the data, adjusting the data to a four parameter logistic curve.

Line 258- About cytotoxicity assay, did you use controls? What were the controls?

As negative control macrophages (without any treatment) were incubated on culture medium.

Results and Discussion

Line 98- I think you can aesthetically improve the Figure 1 by removing the square that externally limits the image as well as the horizontal lines.

This point was corrected in the revised manuscript (pg 3, Figure 1).

Line 107- “…concentration of total phenolic content (Supplementary Materials, Figure S1) . The ethyl acetate…”. Remove space, please!

This point was corrected in the revised manuscript.

Lines 140 and 156- Can you explain exactly what means (-) inactive, (+) low activity, (++) moderate activity, (+++) potent activity? As Alamar Blue® Assay is a quantitative colorimetric assay, can you give the quantitative values?

As suggested by the reviewer, qualitative scale in Tables 2 and 3 is explained in the revised manuscript as follows:

“Key: (-) inactive (viability > 80%), (+) low activity (viability 60-80%), (++) moderate activity (viability 20-60%), (+++) potent activity (viability < 20%).”

Qualitative data were obtained for the extracts and fractions evaluated (microscopy observations). The IC50 values were only calculated for the pure compound. This point was clarified in Materials and Methods section (pg 8, line 261).

153- “…amazonensis and L. donovani, respectively) Moreover, evaluation of this active sub-fraction on murine 153”. Missing punctuation mark.

This point was corrected in the revised manuscript.

Conclusions

Leishmanicidal activity of the thymol derivative was tested on promastigotes that are not the form of the parasite that infect the host cell. However, promastigotes are not the form of the parasite that infects the definitive host. I think it is important to mention this point in the conclusions, stressing that this is a preliminary study. In terms of future prospects, it will be important to explore the potential of P. inuloides and its metabolites in the intracellular amastigote form of the parasite, especially autochthonous Leishmania species.

The aim of our work was to screening the potential of P. inuloides roots for leishmanicidal activity and to identify the compound/compounds responsible for the activity, demonstrating the therapeutic potential of the species under study. The reviewer is right regarding the need to evaluate the identified compound on intracellular amastigote form of parasites, and also to investigate the mode of action of the pure compound. However, this will be the aim of other work.

Moreover, as was underlined in the Conclusions section “Further studies will be conducted in intracellular amastigote form of the parasites to explore the potential of P. inuloides and its metabolites as alternative or complementary leishmanicidal remedies”.

Reviewer 3 Report

I have read with interest the manuscript “Antioxidant and leishmanicidal evaluation of Pulicaria inuloides root extracts: a bioguided  fractionation”

In a previous work, the authors had determined the antiprotozoal activity of extracts from the aerial part of Pulicaria inuloides. In the current study, the authors analyse different activities of root extracts.

This is an interesting and well written manuscript, however, prior to further processing of the paper several points need to be clarified

IN RESULTS AND DISCUSSION SECTION

DPPH radical scavenging activity

-In figure 1, the authors Indicate that the “Data are expressed as mean  ± standard deviation (n =2)”  but  SD is not appreciated properly in the graph

What means N=2?  Is the number of determinations or trials?

Bioassay-guided fractionation

-In table 2 and 3, why don't you use the percentage inhibition for express the results?

-In my opinion a table with IC50, CC50 and selective index of sub-fractions (F1B1, F1B2, F1B3) and miltefosine could be of great help in the interpretation of results.

-I suggest the authors include the leishmanicidal activity of Compound 1 against amastigotes

-The discussion is poor, the authors should include and compare studies related epoxythymol derivatives wih leishmanicidal and/or antioxidant activities.

Author Response

DPPH radical scavenging activity

-In figure 1, the authors Indicate that the “Data are expressed as mean  ± standard deviation (n =2)”  but  SD is not appreciated properly in the graph

What means N=2? Is the number of determinations or trials?

As suggested by the reviewer, Figure 1 has been corrected with the properly SD in the revised manuscript.

N = 2 is the number of determinations as indicated in the Materials and Methods section (pg 7, line 223)

“All determinations were carried out in duplicate”

Bioassay-guided fractionation

-In table 2 and 3, why don't you use the percentage inhibition for express the results?

As suggested by the reviewer, qualitative scale in Tables 2 and 3 is explained in the revised manuscript

-In my opinion a table with IC50, CC50 and selective index of sub-fractions (F1B1, F1B2, F1B3) and miltefosine could be of great help in the interpretation of results.

As suggested by the reviewer a Table (Table 4) with IC50, CC50 and SI of sub-fractions (F1B1, F1B2, F1B3) and miltefosine was included in the revised manuscript.

-I suggest the authors include the leishmanicidal activity of Compound 1 against amastigotes

The aim of our work is to screening the potential of P. inuloides roots for leismanicidal activity and to identify the compound/compounds responsible for activity. The reviewer is right regarding the need to evaluate the identified compound on amastigotes and also to investigate the mode of action. However, this will be the aim of another work.

-The discussion is poor, the authors should include and compare studies related epoxythymol derivatives wih leishmanicidal and/or antioxidant activities.

A paragraph regarding this point has been included in the revised manuscript, and also a reference was added.

“Only about 10% of the known functionalized thymol derivatives have been evaluated as antibacterial, anti-inflammatory, antioxidant, antiprotozoal, cytotoxic, piscicidal, or allelophatic agents. [24]. Regarding epoxythymol derivatives, evaluation on Entamoeba histolytica and Giardia lamblia of thymol isobutyrate derivatives revealed a moderate antiprotozoal activity on both protozoa . However, no study on their leishmanicidal or antioxidant activities has been reported.”

Reference: Bustos-Brito et al Antidiarrheal thymol derivatives from Ageratina glabrata. Structure and absolute configuration of 10-benzoyloxy-8,9-epoxy-6-hydroxythymol isobutyrate. Molecules 2016, 21, 1132.

Round 2

Reviewer 1 Report

The following topics are missing of the revised version:

Tables 2 and 3. Are not convincing. Qualitative data, and explanation of scale, is not strong enough. Data should be presented quantitatively as expressed in Table 4.

EC50 curves of anti-leishmania efficacy and toxicity (table 4) should be shown as part of the manuscript.  

Authors claim that compounds were diluted in DMSO. This is confusing since different solvents and organic extracts were used in the different fractions (lines 222, 223, 301, 302).

Supplementary figure is not shown to reviewer. Cannot view a correlation line between flavonoid content and antioxidant capacity. This figure should be included in the main manuscript (to support the title of publication).

Author Response

Tables 2 and 3. Are not convincing. Qualitative data, and explanation of scale, is not strong enough. Data should be presented quantitatively as expressed in Table 4.

          We agree with the reviewer regarding that quantitative data are more relevant than qualitative one. However, in the present work, and as we previously expose, the first step was the antiparasitic screening of extracts and sub-fractions, and in this sense, we thought was not convenient to spend time and sources until a potential fraction/compound was identified. Thus, qualitative data were obtained for the extracts and fractions evaluated. The IC50 values were only calculated for the pure compound. 

EC50 curves of anti-leishmania efficacy and toxicity (table 4) should be shown as part of the manuscript. 

As suggested by the reviewer we have included the EC50 curves of the anti-Leishmania efficacy of the active compound on L amazonesis and L. donovani and also the EC50 curve of toxicity on macrophages (Figure 2 in the revised manuscript).  

Authors claim that compounds were diluted in DMSO. This is confusing since different solvents and organic extracts were used in the different fractions (lines 222, 223, 301, 302).

The samples were dissolved in DMSO only for the biological assays, and the final DMSO concentration never exceeded 0.1% (v/v) with no effect on the parasites proliferation or morphology. Thus, dry samples (extracts, fractions or pure compound) were dissolved in such solvent to evaluate their activity. On the other hand, the plant material was submitted to different extractor solvents and in this way different extracts were obtained. These extracts were then evaporated, so we had in hand the dry extracts coming from the extraction with different solvents as indicated in the following paragraph:

“Each extract was collected, separately, and concentrated by a rotary vacuum evaporator to remove the solvent to yield the CHCl3 (0.89 g, 0.44% w/w on dry plant material), EtOAc (1 g, 0.50% w/w) and n-BuOH (4 g, 2.0% w/w) extracts.”

Moreover, the active fraction was purified on a preparative silica gel thin layer chromatography (PTLC), which was eluted with mixtures of hexanes-ethyl acetate (50:50) as indicated in the following paragraph:

“The active F1B fraction (36 mg) was purified on a preparative silica gel thin layer chromatography (PTLC), using mixtures of hexanes-ethyl acetate (50:50) to obtain sub-fractions F1B1-F1B3.”  

Supplementary figure is not shown to reviewer. Cannot view a correlation line between flavonoid content and antioxidant capacity. This figure should be included in the main manuscript (to support the title of publication).

The reviewer is right, since a mistake was done in the manuscript concerning a positive correlation between flavonoid content and antioxidant capacity. Since a correlation was observed only for the phenolic content and DPPH radical scavenging. This item has been corrected in the revised manuscript (pgs 3 and 4, lines 131-132).

Reviewer 2 Report

I have no further comments.

Author Response

The Reviewer has no further comments to the revised manuscript.

Round 3

Reviewer 1 Report

Cytotoxicity of the compound to host cells remain as a big issue. New Figure 2 suggest that the compound is highly cytotoxic to host cells. To solve this issue is crucial to claim a compound as potential anti-leishmania agent. Authors need to address cytotoxicity and anti-leishmania activity in an intracellular system (as suggested previously by one of the reviewers). This data is the minimal expected to find in a journal of the quality of Pathogens.

Author Response

REVIEWER1 3º round

Cytotoxicity of the compound to host cells remain as a big issue. New Figure 2 suggest that the compound is highly cytotoxic to host cells. To solve this issue is crucial to claim a compound as potential anti-leishmania agent. Authors need to address cytotoxicity and anti-leishmania activity in an intracellular system (as suggested previously by one of the reviewers). This data is the minimal expected to find in a journal of the quality of Pathogens.

As suggested by the Reviewer we have included the results of the assay of the active compound 1 on L. amazonensis amastigote form.